# Identification of multiple risk loci and regulatory mechanisms influencing susceptibility to multiple myeloma

Molly Went[1], Amit Sud [1], Asta Försti[2,3], Britt-Marie Halvarsson[4], Niels Weinhold et al.[#]

Genome-wide association studies (GWAS) have transformed our understanding of susceptibility to multiple myeloma (MM), but much of the heritability remains unexplained. We report a new GWAS, a meta-analysis with previous GWAS and a replication series, totalling 9974 MM cases and 247,556 controls of European ancestry. Collectively, these data provide evidence for six new MM risk loci, bringing the total number to 23. Integration of information from gene expression, epigenetic profiling and in situ Hi-C data for the 23 risk loci implicate disruption of developmental transcriptional regulators as a basis of MM susceptibility, compatible with altered B-cell differentiation as a key mechanism. Dysregulation of autophagy/apoptosis and cell cycle signalling feature as recurrently perturbed pathways. Our findings provide further insight into the biological basis of MM.

[#]A full list of authors and their affliations appears at the end of the paper.

Multiple myeloma (MM) is a malignancy of plasma cells primarily located within the bone marrow. Although no lifestyle or environmental exposures have been consistently linked to an increased risk of MM, the two- to four-fold increased risk observed in relatives of MM patients provides support for inherited genetic predisposition[1]. Our understanding of MM susceptibility has recently been informed by genome-wide association studies (GWAS), which have so far identified 17 independent risk loci for MM[2–5], with an additional locus being subtype-specific for t(11;14) translocation MM[6]. Much of the heritable risk of MM, however, remains unexplained and statistical modelling indicates that further common risk variants remain to be discovered[7].

To gain a more comprehensive insight into MM aetiology, we performed a new GWAS followed by a meta-analysis with existing GWAS and replication genotyping (totalling 9974 cases and 247,556 controls). Here, we report the identification of six new MM susceptibility loci as well as refined risk estimates for the previously reported loci. In addition, we have investigated the possible gene regulatory mechanisms underlying the associations seen at all 23 GWAS risk loci by analysing in situ promoter Capture Hi-C (CHi-C) in MM cells to characterise chromatin interactions between predisposition single-nucleotide polymorphism (SNPs) and target genes, integrating these data with chromatin immunoprecipitation-sequencing (ChIP-seq) data generated in house and a range of publicly available genomics data. Finally, we have quantified the contribution of both new and previously discovered loci to the heritable risk of MM and implemented a likelihood-based approach to estimate sample sizes required to explain 80% of the heritability.

## Results

**Association analysis**. We conducted a new GWAS using the OncoArray platform[8] (878 MM cases and 7083 controls from the UK), followed by a meta-analysis with six published MM GWAS data sets (totalling 7319 cases and 234,385 controls) (Fig. 1, Supplementary Tables 1–3)[2–5]. To increase genomic resolution, we imputed data to >10 million SNPs. Quantile–quantile (Q–Q) plots for SNPs with minor allele frequency (MAF) >1% after imputation did not show evidence of substantive over-dispersion for the OncoArray GWAS ($\lambda = 1.03$, $\lambda_{1000} = 1.02$, Supplementary Fig. 1). We derived joint odds ratios (ORs) under a fixed-effects model for each SNP with MAF >1%. Finally, we sought validation of nine SNPs associated at $P < 1 \times 10^{-6}$ in the meta-analysis, which did not map to known MM risk loci and displayed a

consistent OR across all GWAS data sets, by genotyping an additional 1777 cases and 6088 controls from three independent series (Germany, Denmark and Sweden). After meta-analysis of the new and pre-existing GWAS data sets and replication series, we identified genome-wide significant associations (i.e. $P < 5 \times 10^{-8}$)[9] for six new loci at 2q31.1, 5q23.2, 7q22.3, 7q31.33, 16p11.2 and 19p13.11 (Table 1, Supplementary Table 4 and 5, Fig. 2). Additionally, borderline associations were identified at two loci with $P$ values of $5.93 \times 10^{-8}$ (6p25.3) and $9.90 \times 10^{-8}$ (7q21.11), which have corresponding Bayesian false-discovery probabilities (BFDP)[10] of 4% and 6%, respectively (Supplementary Table 4 and 5). We found no evidence for significant interactions between any of the 23 risk loci. Finally, we found no evidence to support the existence of the putative risk locus at 2p12.3 (rs1214346), previously proposed by Erickson et al.[11] (GWAS meta-analysis $P$ value = 0.32).

**Risk SNPs and myeloma phenotype**. We did not find any association between sex or age at diagnosis and the 23 MM risk SNPs using case-only analysis (Supplementary Table 6 and 7). Aside from previously reported relationships between the risk loci at 11q13.3 and 5q15 with t(11;14) MM[6] and hyperdiploid MM[12], respectively, we found no evidence for subtype-specific associations (Supplementary Table 8-11) or an impact on MM-specific survival (Supplementary Table 12). A failure to demonstrate additional relationships may, however, be reflective of limited study power. Collectively, these data suggest that the risk variants are likely to have generic effects on MM development.

**Contribution of risk SNPs to heritability**. Using linkage disequilibrium adjusted kinships (LDAK)[13], the heritability of MM ascribable to all common variation was 15.6% (±4.7); collectively the previously identified and new risk loci account for 15.7% of the GWAS heritability (13.6% and 2.1%, respectively). To assess the collective impact of all identified risk SNPs, we constructed polygenic risk scores (PRS) considering the combined effect of all risk SNPs modelled under a log-normal relative risk distribution[14]. Using this approach, an individual in the top 1% of genetic risk has a threefold increased risk of MM when compared to an individual with median genetic risk (Supplementary Fig. 2). We observed an enrichment of risk variants among familial MM compared with both sporadic MM cases and population-based controls comparable to that expected in the absence of a strong monogenic predisposition (respective $P$ values 0.027 and

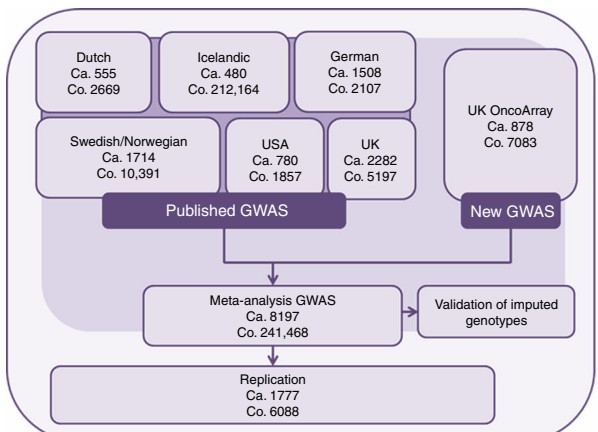
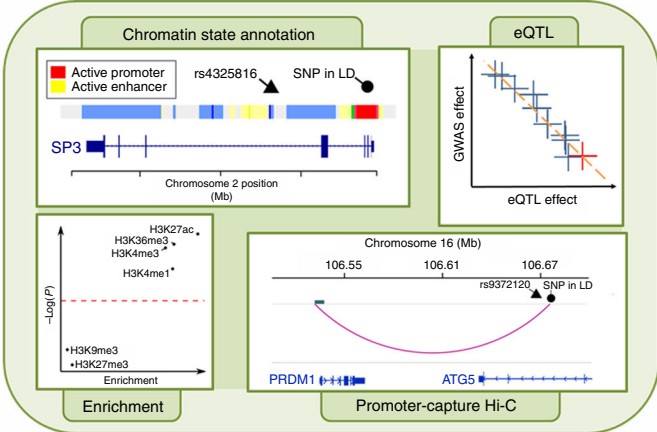

**Fig. 1** GWAS study design. Details of the new and existing GWAS samples, including recruitment centres or trials and quality control, are provided in Supplementary Tables 1 and 2. Trials or centres from which replication samples were recruited are detailed in Supplementary Table 3. Ca.: cases, Co.: controls, eQTL: expression quantitative trait loci, SNP: single-nucleotide polymorphism, LD: linkage disequilibrium

**Table 1 Summary of genotyping results for all 23 risk SNPs**

| SNP | Locus | Pos. (b37) | Risk Allele | RAF | OncoArray | | Previous data | | Replication | | Combined meta | | |
|---|---|---|---|---|---|---|---|---|---|---|---|---|---|
| | | | | | OR | $P_{trend}$ | OR | $P_{trend}$ | OR | $P_{trend}$ | OR | $P_{meta}$ | $I^2$ |
| rs7577599 | 2p23.3 | 25613146 | T | 0.81 | 1.22 | $2.63 \times 10^{-3}$ | 1.24 | $1.24 \times 10^{-16}$ | – | – | 1.23 | $1.29 \times 10^{-18}$ | 0 |
| **rs4325816** | **2q31.1** | **174808899** | **T** | **0.77** | **1.16** | **$1.23 \times 10^{-2}$** | **1.11** | **$1.30 \times 10^{-5}$** | **1.16** | **$3.00 \times 10^{-3}$** | **1.12** | **$7.37 \times 10^{-9}$** | **9** |
| rs6599192 | 3p22.1 | 41992408 | G | 0.16 | 1.24 | $1.35 \times 10^{-3}$ | 1.26 | $8.75 \times 10^{-18}$ | – | – | 1.26 | $4.96 \times 10^{-20}$ | 0 |
| rs10936600 | 3q26.2 | 169514585 | A | 0.75 | 1.18 | $5.12 \times 10^{-3}$ | 1.20 | $5.94 \times 10^{-15}$ | – | – | 1.20 | $1.20 \times 10^{-16}$ | 0 |
| rs1423269 | 5q15 | 95255724 | A | 0.75 | 1.09 | 0.125 | 1.17 | $1.57 \times 10^{-11}$ | – | – | 1.16 | $8.30 \times 10^{-12}$ | 23 |
| **rs6595443** | **5q23.2** | **122743325** | **T** | **0.43** | **1.14** | **$9.87 \times 10^{-3}$** | **1.10** | **$4.69 \times 10^{-6}$** | **1.10** | **0.022** | **1.11** | **$1.20 \times 10^{-8}$** | **0** |
| rs34229995 | 6p22.3 | 15244018 | G | 0.02 | 1.05 | 0.781 | 1.40 | $1.76 \times 10^{-8}$ | – | – | 1.36 | $5.60 \times 10^{-8}$ | 0 |
| rs3132535 | 6p21.3 | 31116526 | A | 0.29 | 1.26 | $2.67 \times 10^{-5}$ | 1.20 | $2.97 \times 10^{-17}$ | – | – | 1.21 | $6.00 \times 10^{-21}$ | 0 |
| rs9372120 | 6q21 | 106667535 | G | 0.21 | 1.18 | $7.74 \times 10^{-3}$ | 1.20 | $8.72 \times 10^{-14}$ | – | – | 1.19 | $2.40 \times 10^{-15}$ | 0 |
| rs4487645 | 7p15.3 | 21938240 | C | 0.65 | 1.23 | $1.06 \times 10^{-4}$ | 1.24 | $5.30 \times 10^{-25}$ | – | – | 1.24 | $2.80 \times 10^{-28}$ | 0 |
| **rs17507636** | **7q22.3** | **106291118** | **C** | **0.74** | **1.12** | **$5.71 \times 10^{-2}$** | **1.12** | **$5.54 \times 10^{-7}$** | **1.10** | **0.036** | **1.12** | **$9.20 \times 10^{-9}$** | **50** |
| **rs58618031** | **7q31.33** | **124583896** | **T** | **0.72** | **1.17** | **$7.61 \times 10^{-3}$** | **1.11** | **$4.70 \times 10^{-6}$** | **1.10** | **0.061** | **1.12** | **$2.73 \times 10^{-8}$** | **0** |
| rs7781265 | 7q36.1 | 150950940 | A | 0.12 | 1.33 | $3.23 \times 10^{-4}$ | 1.20 | $1.82 \times 10^{-7}$ | – | – | 1.22 | $4.82 \times 10^{-10}$ | 49 |
| rs1948915 | 8q24.21 | 128222421 | C | 0.32 | 1.19 | $1.68 \times 10^{-3}$ | 1.14 | $3.14 \times 10^{-10}$ | – | – | 1.15 | $2.53 \times 10^{-12}$ | 26 |
| rs2811710 | 9p21.3 | 21991923 | C | 0.63 | 1.13 | $1.76 \times 10^{-2}$ | 1.14 | $6.50 \times 10^{-10}$ | – | – | 1.14 | $3.64 \times 10^{-11}$ | 0 |
| rs2790457 | 10p12.1 | 28856819 | G | 0.73 | 1.09 | 0.124 | 1.12 | $8.44 \times 10^{-7}$ | – | – | 1.11 | $2.66 \times 10^{-6}$ | 0 |
| **rs13338946** | **16p11.2** | **30700858** | **C** | **0.26** | **1.17** | **$7.90 \times 10^{-3}$** | **1.12** | **$2.22 \times 10^{-7}$** | **1.26** | **$2.5 \times 10^{-7}$** | **1.15** | **$1.02 \times 10^{-13}$** | **26** |
| rs7193541 | 16q23.1 | 74664743 | T | 0.58 | 1.14 | $9.01 \times 10^{-3}$ | 1.12 | $1.14 \times 10^{-8}$ | – | – | 1.12 | $3.68 \times 10^{-10}$ | 34 |
| rs34562254 | 17p11.2 | 16842991 | A | 0.10 | 1.32 | $7.63 \times 10^{-4}$ | 1.30 | $3.63 \times 10^{-17}$ | – | – | 1.30 | $1.18 \times 10^{-19}$ | 29 |
| **rs11086029** | **19p13.11** | **16438661** | **T** | **0.24** | **1.26** | **$1.02 \times 10^{-4}$** | **1.12** | **$1.69 \times 10^{-6}$** | **1.15** | **$5.00 \times 10^{-3}$** | **1.14** | **$6.79 \times 10^{-11}$** | **42** |
| rs6066835 | 20q13.13 | 47355009 | C | 0.08 | 1.13 | 0.162 | 1.24 | $1.16 \times 10^{-9}$ | – | – | 1.23 | $6.58 \times 10^{-10}$ | 38 |
| rs138747 | 22q13.1 | 35700488 | A | 0.66 | – | – | 1.21 | $2.58 \times 10^{-8}$ | – | – | 1.21 | $2.58 \times 10^{-8}$ | 0 |
| rs139402 | 22q13.1 | 39546145 | C | 0.44 | 1.11 | $4.146 \times 10^{-2}$ | 1.23 | $4.98 \times 10^{-26}$ | – | – | 1.22 | $3.84 \times 10^{-26}$ | 56 |

Newly identified risk loci are emboldened.[1] Where >10 TF were implicated at a locus, only those that overlap with TF which demonstrated enrichment in GM12878 are shown here. A full list of TFs localising to loci are detailed in Supplementary Table 17

$1.60 \times 10^{-5}$; Supplementary Fig. 3). Undoubtedly, the identification of further risk loci through the analysis of larger GWAS are likely to improve the performance of any PRS model. To estimate the sample size required to explain a greater proportion of the GWAS heritability, we implemented a likelihood-based approach using association statistics in combination with LD information to model the effect-size distribution[15,16]. The effect-size distributions for susceptibility SNPs were best modelled using the three-component model (mixture of two normal distributions) (Supplementary Fig. 4). Under this model, to identify SNPs explaining 80% of the GWAS heritability is likely to require sample sizes in excess of 50,000 (Supplementary Fig. 5).

**Functional annotation and biological inference of risk loci.** To the extent that they have been studied, many GWAS risk SNPs localise to non-coding regions and influence gene regulation[17]. To investigate the functional role of previously reported and new MM risk SNPs, we performed a global analysis of SNP associations using ChIP-seq data generated on the MM cell line KMS11 and publicly accessible naive B-cell Blueprint Epigenome Project data[18]. We found enrichment of MM SNPs in regions of active chromatin, as indicated by the presence of H3K27ac, H3K4Me3 and H3K4Me1 marks (Supplementary Fig. 6). We also observed an enrichment of relevant B-cell transcription factor-(TF) binding sites using ENCODE GM12878 lymphoblastoid cell line data (Supplementary Fig. 7). Collectively these data support the tenet that the MM predisposition loci influence risk through effects on *cis*-regulatory networks involved in transcriptional initiation and enhancement.

Since genomic spatial proximity and chromatin looping interactions are key to the regulation of gene expression, we interrogated physical interactions at respective genomic regions in KMS11 and naive B-cells using CHi-C data[19]. We also sought to gain insight into the possible biological mechanisms for associations by performing an expression quantitative trait locus

(eQTL) analysis using mRNA expression data on CD138-purified MM plasma cells; specifically, we used Summary data-based Mendelian Randomisation (SMR) analysis[20] to test for pleiotropy between GWAS signal and *cis*-eQTL for genes within 1 Mb of the sentinel SNP to identify a causal relationship. We additionally annotated risk loci with variants mapping to binding motifs of B-cell-specific TFs. Finally, we catalogued direct promoter variants and non-synonymous coding mutations for genes within risk loci (Supplementary Data 1 and Fig. 1).

Although preliminary, and requiring functional validation, our analysis delineates four potential candidate disease mechanisms across the 23 MM risk loci (Supplementary Data 1). Firstly, four of the risk loci contain candidate genes linked to regulation of cell cycle and genomic instability, as evidenced by Hi-C looping interactions in KMS11 cells to *MTAP* (at 9p21.3) and eQTL effects for *CEP120* (at 5q23.2). *CEP120* is required for microtubule assembly and elongation with overexpression of *CEP120* leading to uncontrolled centriole elongation[21]. rs58618031 (7q31.33) maps 5′ of *POT1*, the protection of telomeres 1 gene. POT1 is part of the shelterin complex that functions to protect telomeres and maintain chromosomal stability[22,23]. While mutated *POT1* is not a feature of MM, it is commonly observed in B-cell chronic lymphocytic leukaemia[24–26]. The looping interaction from the rs58618031 annotated enhancer element implicates *ASB15*. Members of the ASB family feature as protein components of the ubiquitin–proteasome system, intriguingly a therapeutic target in MM[27–29].

Second, candidate genes encoding proteins involved in chromatin remodelling were implicated at three of the MM risk loci, supported by promoter variants at 2q31.1, 7q36.1 and 22q13.1. The new locus at 2q31.1 implicates *SP3*, encoding a TF, which through promoter interaction, has a well-established role in B-cell development influencing the expression of germinal centre genes, including activation-induced cytidine deaminase AID[30,31].

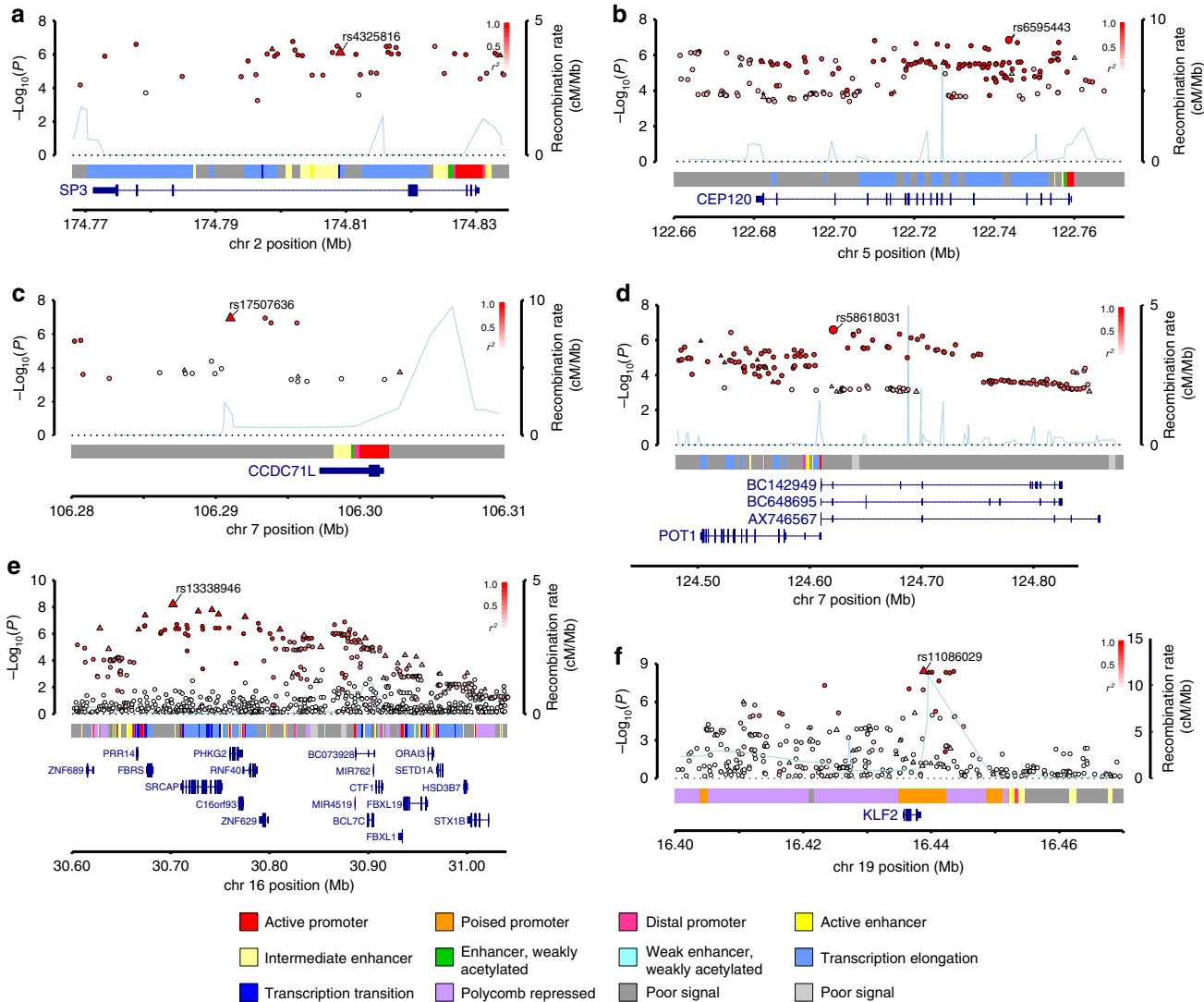

**Fig. 2** Regional plots of the six new risk loci. Regional plots of loci **a** 2q31.1, **b** 5q23.2, **c** 7q22.3, **d** 7q31.33, **e** 16p11.2 and **f** 19p13.11. Plots show results of the meta-analysis for both genotyped (triangles) and imputed (circles) single-nucleotide polymorphisms (SNPs) and recombination rates. $-\log_{10}(P)$ (y axes) of the SNPs are shown according to their chromosomal positions (x axes). The sentinel SNP in each combined analysis is shown as a large circle or triangle and is labelled by its rsID. The colour intensity of each symbol reflects the extent of LD with the top SNP, white ($r^2 = 0$) through to dark red ($r^2 = 1.0$). Genetic recombination rates, estimated using 1000 Genomes Project samples, are shown with a light blue line. Physical positions are based on NCBI build 37 of the human genome. Also shown are the relative positions of genes and transcripts mapping to the region of association. Genes have been redrawn to show their relative positions; therefore maps are not to physical scale. The middle track represents the chromatin-state segmentation track (ChromHMM) for KMS11

Third, the central role *IRF4-MYC*-mediated apoptosis/autophagy in MM oncogenesis is supported by variation at five loci, including eQTL effects *WAC* (at 10p12.1) and Hi-C looping interactions (at 8q24.21 and 16q23.1). The 7p15.3 association ascribable to rs4487645 has been documented to influence expression of *c-MYC*-interacting *CDCA7L* through differential IRF4 binding[32]. Similarly, the long-range interaction between *CCAT1* (colon cancer-associated transcript 1) and *MYC* provides an attractive biological basis for the 8q24.21 association, given the notable role of *MYC* in MM[33,34]. It is noteworthy that the promising risk locus at 6p25.3 contains *IRF4*. At the new locus 19p13.11, the missense variant (NP_057354.1:p.Leu104Pro) and the correlated promoter SNP rs11086029 implicates *KLF2* in MM biology. Demethylation by KDM3A histone demethylase sustains *KLF2* expression and influences IRF4-dependent MM cell survival[35]. The new 16p11.2 risk locus contains a number of genes including Proline-Rich Protein 14 (*PRR14*), which is

implicated in PI3-kinase/Akt/mTOR signalling, a therapeutic target in myelomatous plasma cells[36].

Fourth, loci related to B cell and plasma cell differentiation and function are supported by variation at three loci, including eQTL effects (*ELL2* at 5q15) and Hi-C looping interactions (at 6q21). As previously inferred from GM12878 cell line data, the region at 6q21 (rs9372120, *ATG5*) participates in intra-chromosome looping with the B-cell transcriptional repressor *PRDM1* (alias *BLIMP1*)[4]. Additionally, SNP rs34562254 at 17p11.2 is responsible for the amino acid substitution (NP_036584.1:p.Pro251Leu) in TNFRSF13B, a key regulator of normal B-cell homoeostasis, which has an established role in MM biology[37–42].

## Discussion
Our meta-analysis of a new GWAS series in conjunction with previously published MM data sets has identified six novel risk

loci. Together, the new and previously reported loci explain an estimated 16% of the SNP heritability for MM in European populations. Ancestral differences in the risk of developing MM are well recognised, with a greater prevalence of MM in African Americans as compared with those with European ancestry[43]. It is plausible that the effects of MM risk SNPs may differ between Europeans and non-Europeans and hence contribute to differences in prevalence rates. Thus far, there has only been limited evaluation of this possibility with no evidence for significant differences[44].

Integration of Hi-C data with ChIP-seq chromatin profiling from MM and lymphoblastoid cell lines and naive B cells and eQTL analysis, using patient expression data, has allowed us to gain preliminary insight into the biological basis of MM susceptibility. This analysis suggests a model of MM susceptibility based on transcriptional dysregulation consistent with altered B-cell differentiation, where dysregulation of autophagy/apoptosis and cell cycle signalling feature as recurrently modulated pathways. Specifically, our findings implicate mTOR-related genes *ULK4*, *ATG5* and *WAC*, and by virtue of the role of *IRF4-MYC* related autophagy, *CDCA7L*, *DNMT3A*, *CBX7* and *KLF2* in MM development (Supplementary Data 1). Further investigations are necessary to decipher the functional basis of risk SNPs, nevertheless we highlight mTOR signalling and the ubiquitin–proteasome pathway, targets of approved drugs in MM. As a corollary of this, genes elucidated via the functional annotation of GWAS that discovered MM risk loci may represent promising therapeutic targets for myeloma drug discovery. Finally, our estimation of sample sizes required to identify a larger proportion of the heritable risk of MM attributable to common variation underscore the need for further international collaborative analyses.

## Methods

**Ethics**. Collection of patient samples and associated clinico-pathological information was undertaken with written informed consent and relevant ethical review board approval at respective study centres in accordance with the tenets of the Declaration of Helsinki. Specifically for the Myeloma-IX trial by the Medical Research Council (MRC) Leukaemia Data Monitoring and Ethics committee (MREC 02/8/95, ISRCTN68454111), the Myeloma-XI trial by the Oxfordshire Research Ethics Committee (MREC 17/09/09, ISRCTN49407852), HOVON65/GMMG-HD4 (ISRCTN 644552890; METC 13/01/2015), HOVON87/NMSG18 (EudraCTnr 2007-004007-34, METC 20/11/2008), HOVON95/EMN02 (EudraCTnr 2009-017903-28, METC 04/11/10), University of Heidelberg Ethical Commission (229/2003, S-337/2009, AFmu-119/2010), University of Arkansas for Medical Sciences Institutional Review Board (IRB 202077), Lund University Ethical Review Board (2013/54), the Norwegian REK 2014/97, the Danish Ethical Review Board (no. H-16032570) and Icelandic Data Protection Authority (2,001,010,157 and National Bioethics Committee 01/015).

The diagnosis of MM (ICD-10 C90.0) in all cases was established in accordance with World Health Organization guidelines. All samples from patients for genotyping were obtained before treatment or at presentation.

**Primary GWAS**. We analysed constitutional DNA (EDTA-venous blood derived) from 931 cases ascertained through the UK Myeloma XI trial; detailed in Supplementary Table 1. Cases were genotyped using the Illumina OncoArray (Illumina Inc. San Diego, CA 92122, USA). Controls were also genotyped using the OncoArray and comprised: (1) 2976 cancer-free men recruited by the PRACTICAL Consortium—the UK Genetic Prostate Cancer Study (UKGPCS) (age <65 years), a study conducted through the Royal Marsden NHS Foundation Trust and SEARCH (Study of Epidemiology & Risk Factors in Cancer), recruited via GP practices in East Anglia (2003–2009) and (2) 4446 cancer-free women across the UK, recruited via the Breast Cancer Association Consortium (BCAC).

Standard quality-control measures were applied to the GWAS[45]. Specifically, individuals with low SNP call rate (<95%) as well as individuals evaluated to be of non-European ancestry (using the HapMap version 2 CEU, JPT/CHB and YRI populations as a reference) were excluded (Supplementary Fig. 8). For apparent first-degree relative pairs, we excluded the control from a case–control pair; otherwise, we excluded the individual with the lower call rate. SNPs with a call rate <95% were excluded as were those with a MAF <0.01 or displaying significant deviation from Hardy–Weinberg equilibrium ($P < 10^{-5}$). GWAS data were imputed to >10 million SNPs using IMPUTE2 v2.3[46] software in conjunction with

a merged reference panel consisting of data from 1000 Genomes Project[47] (phase 1 integrated release 3 March 2012) and UK10K[48]. Genotypes were aligned to the positive strand in both imputation and genotyping. We imposed predefined thresholds for imputation quality to retain potential risk variants with MAF >0.01 for validation. Poorly imputed SNPs with an information measure <0.80 were excluded. Tests of association between imputed SNPs and MM was performed under an additive model in SNPTESTv2.5[49]. The adequacy of the case–control matching and possibility of differential genotyping of cases and controls was evaluated using a Q–Q plot of test statistics (Supplementary Fig. 1). The inflation $\lambda$ was based on the 90% least-significant SNPs[50] and assessment of $\lambda_{1000}$. Details of SNP QC are provided in in Supplementary Table 2.

**Published GWAS**. The data from six previously reported GWAS[2–5] are summarised in Supplementary Table 1. All these studies were based on individuals with European ancestry and comprised: UK-GWAS (2282 cases, 5197 controls), Swedish-GWAS (1714 cases, 10,391 controls), German-GWAS (1508 cases, 2107 controls), Netherlands-GWAS (555 cases, 2669 controls), US-GWAS (780 cases, 1857 controls) and Iceland (480 cases, 212,164 controls).

**Replication studies and technical validation**. To validate promising associations, we analysed three case–control series from Germany, Sweden and Denmark, summarised in Supplementary Table 3. The German replication series comprised 911 cases collected by the German Myeloma Study Group (Deutsche Studiengruppe Multiples Myeloma (DSMM)), GMMG, University Clinic, Heidelberg and University Clinic, Ulm. Controls comprised 1477 healthy German blood donors recruited between 2004 and 2007 by the Institute of Transfusion Medicine and Immunology, University of Mannheim, Germany. The Swedish replication series comprised 534 MM cases from the Swedish National Myeloma Biobank and the Danish replication series comprised 332 MM cases from the University Hospital of Copenhagen. As controls, we analysed 2382 Swedish blood donors and 2229 individuals from Denmark and Skåne County, Sweden (the southernmost part of Sweden adjacent to Denmark). Replication genotyping of German and Scandinavian samples was performed using competitive allele-specific PCR KASPar chemistry (LGC, Hertfordshire, UK). Call rates for SNP genotypes were >95% in each of the replication series. To ensure the quality of genotyping in all assays, at least two negative controls and duplicate samples (showing a concordance of >99%) were genotyped at each centre. The fidelity of imputation was assessed by directly sequencing a set of 147 randomly selected samples from the UK OncoArray case series. Imputation was found to be robust; concordance was >90% (Supplementary Table 13). Genotyping and sequencing primers are detailed in Supplementary Table 14 and 15, respectively.

**Meta-analysis**. Meta-analyses were performed using the fixed-effects inverse-variance method using META v1.6[51]. Cochran's Q-statistic to test for heterogeneity and the $I^2$ statistic to quantify the proportion of the total variation due to heterogeneity was calculated. Using the meta-analysis summary statistics and LD correlations from a reference panel of the 1000 Genomes Project combined with UK10K, we implemented Genome-wide Complex Trait Analysis[52] to perform conditional association analysis. Association statistics were calculated for all SNPs conditioning on the top SNP in each loci showing genome-wide significance. This was carried out step-wise.

For borderline associations, the BFDP[10] was calculated based on a plausible OR of 1.2 and a prior probability of association of 0.0001. For both promising associations, the BFDP was <10%.

**Fluorescence in situ hybridisation**. Fluorescence in situ hybridisation (FISH) and ploidy classification of UK and German samples were performed as previously described[53,54]. Logistic regression in case-only analyses was used to assess the relationship between SNP genotype and IgH translocations or tumour ploidy.

**eQTL analysis**. eQTL analyses were performed using CD138-purified plasma cells from 183 UK MyIX trial patients and 658 German GMMG patients[32]. Briefly, German and UK data were pre-processed separately, followed by analysis using a Bayesian approach to probabilistic estimation of expression residuals to infer broad variance components, accounting for hidden determinants influencing global expression. The association between genotype of SNPs and expression of genes within 1 Mb either side of each MM risk locus was evaluated based on the significance of linear regression coefficients. We pooled data from each study under a fixed-effects model.

The relationship between SNP genotype and gene expression was carried out using SMR analysis as per Zhu et al.[2] Briefly, if $b_{xy}$ is the effect size of $x$ (gene expression) on $y$ (slope of $y$ regressed on the genetic value of $x$), $b_{zx}$ is the effect of $z$ on $x$ and $b_{zy}$ be the effect of $z$ on $y$, $b_{xy}$ ($b_{zy}/b_{zx}$) is the effect of $x$ on $y$. To distinguish pleiotropy from linkage where the top associated *cis*-eQTL is in LD with two causal variants, one affecting gene expression and the other affecting a trait, we tested for heterogeneity in dependent instruments (HEIDI), using multiple SNPs in each *cis*-eQTL region. Under the hypothesis of pleiotropy, $b_{xy}$ values for SNPs in LD with the causal variant should be identical. For each probe that passed significance

threshold for the SMR test, we tested the heterogeneity in the $b_{xy}$ values estimated for multiple SNPs in the *cis*-eQTL region using HEIDI.

GWAS summary statistics files were generated from the meta-analysis. We set a threshold for the SMR test of $P_{SMR} < 1 \times 10^{-3}$ corresponding to a Bonferroni correction for 45 tests, i.e. 45 probes which demonstrated an association in the SMR test. For all genes passing this threshold, we generated plots of the eQTL and GWAS associations at the locus, as well as plots of GWAS and eQTL effect sizes (i.e. input for the HEIDI heterogeneity test). HEIDI test $P$ values < 0.05 were considered as reflective of heterogeneity. This threshold is, however, conservative for gene discovery because it retains fewer genes than when correcting for multiple testing. SMR plots for significant eQTLs are shown in Supplementary Fig. 9 and 10, and a summary of results are shown in Supplementary Table 16.

**Promoter CHi-C.** To map risk SNPs to interactions involving promoter contacts and identify genes involved in MM susceptibility, we analysed publicly accessible promoter CHi-C data on the naive B cells downloaded from Blueprint Epigenome Project. Additionally, we also analysed promoter CHi-C data that we have previously generated for the MM cell line KMS11[12]. Interactions were called using the CHiCAGO pipeline to obtain a unique list of reproducible contacts[55] and those with a −log(weighted $P$) ≥5 were considered significant.

**Chromatin state annotation.** Variant sets (i.e. sentinel risk SNP and correlated SNPs, $r^2 > 0.8$) were annotated for putative functional effect based upon histone mark ChIP-seq data for H3K27ac, H3K4Me1, H3K27Me3, H3K9Me3, H3K36Me3 and H3K27Me3 from KMS11 cell lines, generated in-house and naive B cells from Blueprint Epigenome Project[56]. We used ChromHMM to infer chromatin states by integrating information on these histone modifications, training the model on three MM cell lines; KMS11, MM1S and JJN3. Genome-wide signal tracks were binarized (including input controls for ChIP-seq data), and a set of learned models were generated using ChromHMM software[57]. A 12-state model was suitable for interpretation, and biological meaning was assigned to the states based on chromatin marks that use putative rules as previously described (Supplementary Fig. 11).

**TF and histone mark enrichment analysis.** To examine enrichment in specific TF binding across risk loci, we adapted the method of Cowper-Sal lari et al.[58]. Briefly, for each risk locus, a region of strong LD (defined as $r^2 > 0.8$ and $D' > 0.8$) was determined, and these SNPs were considered the associated variant set (AVS). Publically available data on TF ChIP-seq uniform peak data were obtained from ENCODE for the GM12878 cell line, including data for 82 TF and 11 histone marks[59]. In addition, ChIP-seq peak data for six histone marks from KMS11 cell line were generated in-house, and naive B-cell ChIP-seq data were downloaded from Blueprint Epigenome Project[56]. For each mark, the overlap of the SNPs in the AVS and the binding sites was assessed to generate a mapping tally. A null distribution was produced by randomly selecting SNPs with the same characteristics as the risk-associated SNPs, and the null mapping tally calculated. This process was repeated 10,000 times, and $P$ values were calculated as the proportion of permutations, where null mapping tally was greater or equal to the AVS mapping tally. An enrichment score was calculated by normalising the tallies to the median of the null distribution. Thus, the enrichment score is the number of standard deviations of the AVS mapping tally from the median of the null distribution tallies. Enrichment plots are shown in Supplementary Fig. 6 and 7.

**Functional annotation.** For the integrated functional annotation of risk loci, variant sets (i.e. all SNPs in LD $r^2 > 0.8$ with the sentinel SNP) were annotated with: (i) presence of a Hi-C contact linking to a gene promoter, (ii) presence of an association from SMR analysis, (iii) presence of a regulatory ChromHMM state, (iv) evidence of transcription factor binding and (v) presence of a nonsynonymous coding change. Candidate causal genes were then assigned to MM risk loci using the target genes implicated in annotation tracks (i), (ii), (iiii) and (iv). If the data supported multiple gene candidates, the gene with the highest number of individual functional data points was considered as the candidate. Where multiple genes have the same number of data points, all genes are listed. Direct non-synonymous coding variants were allocated additional weighting. Competing mechanisms for the same gene (e.g. both coding and promoter variants) were permitted.

**Heritability analysis.** We used LDAK to estimate the polygenic variance (i.e. heritability) ascribable to all genotyped and imputed GWAS SNPs from summary statistic data. SNP-specific expected heritability, adjusted for LD, MAF and genotype certainty was calculated from the UK10K and 1000 Genomes data. Samples were excluded with a call rate <0.99 or if individuals were closely related or of divergent ancestry from CEU. Individual SNPs were excluded if they showed deviation from HWE with $P < 1 \times 10^{-5}$, an individual SNP genotype yield <95%, MAF <1%, SNP imputation score <0.99 and the absence of the SNP in the GWAS summary statistic data. This resulted in a total 1,254,459 SNPs which were used to estimate the heritability of MM.

To estimate the sample size required for a given proportion of the GWAS heritability, we implemented a likelihood-based approach to model the effect-size distribution[15] using association statistics, from the MM meta-analysis, and LD information, obtained from individuals of European ancestry in the 1000 Genomes

Project Phase 3. LD values were based on an $r^2$ threshold of 0.1 and a window size of 1 MB. The goodness of fit of the observed distribution of $P$ values against the expected from a two-component model (single normal distribution) and a three-component model (mixture of two normal distributions) were assessed[15], and a better fit was observed for the latter model (Supplementary Figure 4). The percentage of GWAS heritability explained for a projected sample size was determined using this model and is based on power calculations for the discovery of genome-wide significant SNPs. The genetic variance explained was calculated as the proportion of total GWAS heritability explained by SNPs reaching genome-wide significance at a given sample size. The 95% confidence intervals were determined using 10,000 simulations.

PRS for familial MM ($n = 38$) from 25 families were compared with sporadic MM ($n = 1530$) and population-based controls ($n = 10,171$); first as a simple sum of risk alleles and secondly as sum of risk alleles weighted by their log-transformed ORs. Family member scores were averaged. A one-sided Student's *t*-test was used to assess difference between groups. The genetic data have been previously described[5,60] with the familial MM cases having been identified by linkages of Swedish registry information.

**Data availability.** SNP genotyping data that support the findings of this study have been deposited in Gene Expression Omnibus with accession codes GSE21349, GSE19784, GSE24080, GSE2658 and GSE15695; in the European Genome-phenome Archive (EGA) with accession code EGAS00000000001; in the European Bioinformatics Institute (Part of the European Molecular Biology Laboratory) (EMBL-EBI) with accession code E-MTAB-362 and E-TABM-1138; and in the database of Genotypes and Phenotypes (dbGaP) with accession code phs000207.v1. p1. Expression data that support the findings of this study have been deposited in GEO with accession codes GSE21349, GSE2658, GSE31161 and EMBL-EBI with accession code E-MTAB-2299. The remaining data are contained within the paper and Supplementary Files or are available from the author upon request. KMS11 Hi-C data used in this manuscript are deposited in EGA under accession number EGAS00001002614. The accession number for the KMS11 ChIP-seq data reported in this paper is EGA: S00001002414. Naive B-cell HiC data used in this work is publically available from Blueprint Blueprint Epigenome Project [https://osf.io/u8tzp/]. ChIP-seq data for H3K27ac, H3K4Me1, H3K27Me3, H3K9Me3, H3K36Me3 and H3K27Me3 from naive B cells are publically available and were obtained from Blueprint Epigenome Project [http://www.blueprint-epigenome.eu/].

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

## Acknowledgements

In the United Kingdom, Myeloma UK and Bloodwise provided principal funding. Additional funding was provided by Cancer Research UK (C1298/A8362 supported by the Bobby Moore Fund) and The Rosetrees Trust. M.W. is supported by funding from Mr Ralph Stockwell. A.S. is supported by a clinical fellowship from Cancer Research UK and charitable funds from the Royal Marsden Hospital. N.W. was supported by the National Institute of General Medical Sciences of the National Institutes of Health under Award Number P20GM125503. This study made use of genotyping data on the 1958 Birth Cohort generated by the Wellcome Trust Sanger Institute (http://www.wtccc.org.uk). We thank the High-Throughput Genomics Group at the Wellcome Trust Centre for Human Genetics (funded by Wellcome Trust grant reference 090532/Z/09/Z) for the generation of UK myeloma OncoArray data. The BCAC study would not have been possible without the contributions of the following: Manjeet K. Bolla, Qin Wang, Kyriaki Michailidou and Joe Dennis. BCAC is funded by Cancer Research UK (C1287/A10118, C1287/A16563). For the BBCS study, we thank Eileen Williams, Elaine Ryder-Mills and Kara Sargus. The BBCS is funded by Cancer Research UK and Breast Cancer Now and acknowledges NHS funding to the National Institute of Health Research (NIHR) Bio-medical Research Centre (BRC), and the National Cancer Research Network (NCRN). We thank the participants and the investigators of EPIC (European Prospective Investigation into Cancer and Nutrition). The coordination of EPIC is financially supported by the European Commission (DG-SANCO) and the International Agency for Research on Cancer. The national cohorts are supported by: Ligue Contre le Cancer, Institut Gustave Roussy, Mutuelle Générale de l'Education Nationale, Institut National de la Santé et de la Recherche Médicale (INSERM) (France); German Cancer Aid, German Cancer Research Center (DKFZ), Federal Ministry of Education and Research (BMBF) (Germany); the Hellenic Health Foundation, the Stavros Niarchos Foundation (Greece); Associazione Italiana per la Ricerca sul Cancro-AIRC-Italy and National Research Council (Italy); Dutch Ministry of Public Health, Welfare and Sports (VWS), Netherlands Cancer Registry (NKR), LK Research Funds, Dutch Prevention Funds, Dutch ZON (Zorg Onderzoek Nederland), World Cancer Research Fund (WCRF), Statistics Netherlands (The Netherlands); Health Research Fund (FIS), PI13/00061 to Granada, PI13/01162 to EPIC-Murcia, Regional Governments of Andalucía, Asturias, Basque Country, Murcia and Navarra, ISCIII RETIC (RD06/0020) (Spain); Cancer Research UK (14136 to EPIC-Norfolk; C570/A16491 and C8221/A19170 to EPIC-Oxford), Medical Research Council (1000143 to EPIC-Norfolk, MR/M012190/1 to EPIC-Oxford) (UK). We thank the SEARCH and EPIC teams which were funded by a programme grant from Cancer Research UK [A16561] and supported by the UK NIHR BRC at the University of Cambridge. We thank Breast Cancer Now and The Institute of Cancer Research (ICR) for support and funding of the UKBGS, and the study participants, study staff, and the

doctors, nurses and other health care providers and health information sources who have contributed to the study. We acknowledge NHS funding to the Royal Marsden/ICR NIHR BRC. UKGPCS would like to thank The Institute of Cancer Research and The Everyman Campaign for funding support. The UKGPCS acknowledges The Prostate Cancer Research Foundation, Prostate Action, The Orchid Cancer Appeal, The National Cancer Research Network UK, The National Cancer Research Institute (NCRI), the NIHR funding to the NIHR Biomedical Research data managers and consultants for their work in the UKGPCS study and urologists and other persons involved in the planning, and data collection of the CAPS study. Genotyping of the OncoArray was funded by the US National Institutes of Health (NIH) [U19 CA 148537 for ELucidating Loci Involved in Prostate cancer SuscEptibility (ELLIPSE) project and X01HG007492 to the Center for Inherited Disease Research (CIDR) under contract number HHSN268201200008I]. Additional analytic support was provided by NIH NCI U01 CA188392 (PI: Schumacher). The PRACTICAL consortium was supported by Cancer Research UK Grants C5047/A7357, C1287/A10118, C1287/A16563, C5047/A3354, C5047/A10692, C16913/A6135, European Commission's Seventh Framework Programme grant agreement no. 223175 (HEALTH-F2-2009-223175) and The National Institute of Health (NIH) Cancer Post-Cancer GWAS initiative grant no. 1 U19 CA 148537-01 (the GAME-ON initiative). We would also like to thank the following for funding support: The Institute of Cancer Research and The Everyman Campaign, The Prostate Cancer Research Foundation, Prostate Research Campaign UK (now Prostate Action), The Orchid Cancer Appeal, The National Cancer Research Network UK, The National Cancer Research Institute (NCRI) UK. We are grateful for support of NIHR funding to the NIHR Biomedical Research Centre at The Institute of Cancer Research and The Royal Marsden NHS Foundation Trust. The APBC BioResource, which forms part of the PRACTICAL consortium, consists of the following members: Wayne Tilley, Gail Risbridger, Renea Taylor, Judith A Clements, Lisa Horvath, Vanessa Hayes, Lisa Butler, Trina Yeadon, Allison Eckert, Pamela Saunders, Anne-Maree Haynes, Melissa Papargiris. We thank the staff of the CTRU University of Leeds and the NCRI haematology Clinical Studies Group. The US GWAS was supported by a grant from the National Institutes of Health (P01CA055819). The German study was supported by the Dietmar-Hopp-Stiftung, Germany, the German Cancer Aid (110,131), the German Ministry of Education and Science (CLIOMMICS 01ZX1309), The German Research Council (DFG; Project SI 236/81, SI 236/)-1, ER 155/6-1 and the DFG CRI 216), the Harald Huppert Foundation and the Multiple Myeloma Research Foundation. The patients were collected by the GMMG and DSMM studies. The German GWAS made use of genotyping data from the population-based HNR study, which is supported by the Heinz Nixdorf Foundation (Germany). The genotyping of the Illumina HumanOmni-1 Quad BeadChips of the HNR subjects was financed by the German Center for Neurodegenerative Disorders (DZNE), Bonn. We are grateful to all investigators who contributed to the generation of this data set. The German replication controls were collected by Peter Bugert, Institute of Transfusion Medicine and Immunology, Medical Faculty Mannheim, Heidelberg University, German Red Cross Blood Service of Baden-Württemberg-Hessen, Mannheim, Germany. This work was supported by research grants from the Swedish Foundation for Strategic Research (KF10-0009), the Marianne and Marcus Wallenberg Foundation (2010.0112), the Knut and Alice Wallenberg Foundation (2012.0193), the Swedish Research Council (2012–1753), the Royal Swedish Academy of Science, ALF grants to the University and Regional Laboratories (Labmedicin Skåne), the Siv-Inger and Per-Erik Andersson Foundation, the Medical Faculty at Lund University, the Borås foundation for cancer research, and the Swedish Society of Medicine. We thank Jörgen Adolfsson, Tomas Axelsson, Anna Collin, Ildikó Frigyesi, Patrik Magnusson, Bertil Johansson, Jan Westin and Helga Ögmunds-dóttir for their assistance. This work was supported by Center for Translational Molecular Medicine (BioCHIP), a clinical research grant from the European Hematology Association, an EMCR Translational Research Grant, a BMBF grant from CLIOMMICS (01ZX1309A) and FP7 grant MSCNET (LSHC-Ct-2006-037602). We thank the staff of the HOVON, as well as patients and physicians at participating sites. In addition, we also thank Jasper Koenders, Michael Vermeulen, André Uitterlinden and Nathalie van der Velde for their assistance. We are indebted to the clinicians who contributed samples to Swedish, Icelandic, Norwegian and Danish biobanks. We are indebted to the patients and other individuals who participated in the project.

## Author contributions

M.W. and R.S.H. designed the study. M.W. and R.S.H. drafted the manuscript with contributions from K.H., B.N., A.S. and N.W. M.W. performed principal statistical and bioinformatics analyses. A.S., N.L., P.L., D.C.J., J.S.M. and G.O. performed additional bioinformatics analyses. S.K. performed in situ CHi-C. P.B. coordinated UK laboratory analyses. M.W. and A.H. performed sequencing of UK samples. D.C.J. managed and prepared Myeloma IX and Myeloma XI Case Study DNA samples. M.K., G.J.M., F.E.D., W.A.G. and G.H.J. performed ascertainment and collection of Case Study samples. B.A.W. performed UK expression analyses. F.M.R. performed UK fluorescence in situ hybridisation analyses. The PRACTICAL consortium, D.E., P.P., A.D., J.P., F.C., R.E., Z.K.-J., K.M. and N.P. provided control samples for the UK OncoArray GWAS. H.G., U.B., J.H., J.N. and N.W. coordinated and managed Heidelberg samples. C.L. and H.E. coordinated and managed Ulm/Wurzburg samples. A.F. coordinated German genotyping. C.C. and O.R.B. performed German genotyping. P.H. and M.M.N. performed GWAS of German cases and controls. H.T., B.C. and M.I.d.S.F. carried out statistical analysis. K.H. coordinated the German part of the project. M.M.N. generated genotype data from the Heinz-Nixdorf recall study. K.-H.J. contributed towards the Heinz-Nixdorf control data set. N.N. from Bonn and K.-H.J. provided samples for the German GWAS. M.H. and B.N. coordinated the Swedish/Norwegian part of the project. M.A. and B.-M. H. performed data analysis. E.J., A.-K.W., U.-H.M., H.N., A.V., N.F.A., A.W., I.T. and U.G. performed sample acquisition, sample preparation, clinical data acquisition and additional data analyses of Sweden/Norway samples. In Iceland, G.T. and D.F.G. performed statistical analysis. S.Y.K. provided clinical information. T.R. performed additional analyses. U.T. and K.S. performed project oversight. M.v.D., P.S., A.B. and R.K. coordinated HOVON65/GMMG-HD4, HOVON87/NMSG18 and HOVON95/EMN02 studies for participating in this study, and coordinated genotyping and pre-processing. At the Myeloma Institute, University of Arkansas for Medical Sciences, N.W. coordinated the US part of the project and performed statistical and eQTL analyses. O.W.S. and N.W. managed Case Study samples. G.J.M. and F.E.D. performed ascertainment and collection of Case Study samples.

## Additional information

**Competing interests:** G.T., D.F.G., T.R., K.S. and U.T. are employed by deCode Genetics/Amgen Inc. The remaining authors declare no competing interests.

Molly Went[1], Amit Sud[1], Asta Försti[2,3], Britt-Marie Halvarsson[4], Niels Weinhold[5,6], Scott Kimber[7], Mark van Duin[8], Gudmar Thorleifsson[9], Amy Holroyd[1], David C. Johnson[7], Ni Li[1], Giulia Orlando[1], Philip J. Law[1], Mina Ali[4], Bowang Chen[2], Jonathan S. Mitchell[1], Daniel F. Gudbjartsson[9,10], Rowan Kuiper[8], Owen W. Stephens[5], Uta Bertsch[2,11], Peter Broderick[1], Chiara Campo[2], Obul R Bandapalli[2], Hermann Einsele[12], Walter A. Gregory[13], Urban Gullberg[4], Jens Hillengass[6], Per Hoffmann[14,15], Graham H. Jackson[16], Karl-Heinz Jöckel[17], Ellinor Johnsson[4], Sigurður Y. Kristinsson[18], Ulf-Henrik Mellqvist[19], Hareth Nahi[20], Douglas Easton[21,22], Paul Pharoah[21,22], Alison Dunning[21], Julian Peto[23],

Federico Canzian [24], Anthony Swerdlow[1,25], Rosalind A. Eeles [1,26], ZSofia Kote-Jarai[1], Kenneth Muir [27,28], Nora Pashayan [21,29], The PRACTICAL consortium, Jolanta Nickel[6], Markus M. Nöthen[14,30], Thorunn Rafnar[9], Fiona M. Ross[31], Miguel Inacio da Silva Filho[2], Hauke Thomsen[2], Ingemar Turesson[32], Annette Vangsted[33], Niels Frost Andersen[34], Anders Waage[35], Brian A. Walker[5], Anna-Karin Wihlborg[4], Annemiek Broyl[8], Faith E. Davies[5], Unnur Thorsteinsdottir[9,36], Christian Langer[37], Markus Hansson [4,32], Hartmut Goldschmidt[6,11], Martin Kaiser[7], Pieter Sonneveld[8], Kari Stefansson[9], Gareth J. Morgan[5], Kari Hemminki[2,3], Björn Nilsson[4,38] & Richard S. Houlston [1,7]

[1]Division of Genetics and Epidemiology, The Institute of Cancer Research, London SW7 3RP, UK. [2]German Cancer Research Center, 69120, Heidelberg, Germany. [3]Center for Primary Health Care Research, Lund University, SE-205 02, Malmo, Sweden. [4]Hematology and Transfusion Medicine, Department of Laboratory Medicine, BMC B13, Lund University, SE-221 84, Lund, Sweden. [5]Myeloma Institute for Research and Therapy, University of Arkansas for Medical Sciences, Little Rock, AR 72205, USA. [6]Department of Internal Medicine V, University of Heidelberg, 69117, Heidelberg, Germany. [7]Division of Molecular Pathology, The Institute of Cancer Research, London SW7 3RP, UK. [8]Department of Hematology, Erasmus MC Cancer Institute, 3075 EA, Rotterdam, The Netherlands. [9]deCODE Genetics, Sturlugata 8, IS-101, Reykjavik, Iceland. [10]School of Engineering and Natural Sciences, University of Iceland, IS-101, Reykjavik, Iceland. [11]National Centre of Tumor Diseases, 69120, Heidelberg, Germany. [12]University Clinic of Würzburg, 97080, Würzburg, Germany. [13]Clinical Trials Research Unit, University of Leeds, Leeds LS2 9PH, UK. [14]Institute of Human Genetics, University of Bonn, D-53127, Bonn, Germany. [15]Division of Medical Genetics, Department of Biomedicine, University of Basel, 4003, Basel, Switzerland. [16]Royal Victoria Infirmary, Newcastle upon Tyne NE1 4LP, UK. [17]Institute for Medical Informatics, Biometry and Epidemiology, University Hospital Essen, University of Duisburg–Essen, Essen D-45147, Germany. [18]Department of Hematology, Landspitali, National University Hospital of Iceland, IS-101, Reykjavik, Iceland. [19]Section of Hematology, Sahlgrenska University Hospital, Gothenburg 413 45, Sweden. [20]Center for Hematology and Regenerative Medicine, SE-171 77, Stockholm, Sweden. [21]Centre for Cancer Genetic Epidemiology, Department of Oncology, University of Cambridge, Cambridge CB1 8RN, UK. [22]Centre for Cancer Genetic Epidemiology, Department of Public Health and Primary Care, University of Cambridge, Cambridge CB1 8RN, UK. [23]Department of Non-Communicable Disease Epidemiology, London School of Hygiene and Tropical Medicine, London WC1E 7HT, UK. [24]Genomic Epidemiology Group, German Cancer Research Center (DKFZ), Heidelberg 69120, Germany. [25]Division of Breast Cancer Research, The Institute of Cancer Research, London SW7 3RP, UK. [26]Royal Marsden NHS Foundation Trust, Fulham Road, London SW3 6JJ, UK. [27]Institute of Population Health, University of Manchester, Manchester M13 9PL, UK. [28]Warwick Medical School, University of Warwick, Coventry CV4 7AL, UK. [29]Department of Applied Health Research, University College London, London WC1E 7HB, UK. [30]Department of Genomics, Life & Brain Center, University of Bonn, D-53127, Bonn, Germany. [31]Wessex Regional Genetics Laboratory, University of Southampton, Salisbury SP2 8BJ, UK. [32]Hematology Clinic, Skåne University Hospital, SE-221 85, Lund, Sweden. [33]Department of Haematology, University Hospital of Copenhagen at Rigshospitalet, Blegdamsvej 9, DK-2100, Copenhagen, Denmark. [34]Department of Haematology, Aarhus University Hospital, Tage-Hansens Gade 2, DK-8000, Aarhus C, Denmark. [35]Department of Cancer Research and Molecular Medicine, Norwegian University of Science and Technology, Box 8905, N-7491, Trondheim, Norway. [36]Faculty of Medicine, University of Iceland, IS-101, Reykjavik, Iceland. [37]Department of Internal Medicine III, University of Ulm, D-89081, Ulm, Germany. [38]Broad Institute, 7 Cambridge Center, Cambridge, MA 02142, USA. These authors contributed equally: Molly Went, Amit Sud, Asta Försti, Britt-Marie Halvarsson, Niels Weinhold. These authors jointly supervised the work: Hartmut Goldschmidt, Kari Stefansson, Gareth J. Morgan, Björn Nilsson, Kari Hemminki, Richard S. Houlston.

## The PRACTICAL consortium

Brian E. Henderson[39], Christopher A. Haiman[39], Sara Benlloch[1], Fredrick R. Schumacher[40,41], Ali Amin Al Olama[22,42], Sonja I. Berndt[43], David V. Conti[39], Fredrik Wiklund[44], Stephen Chanock[43], Victoria L. Stevens[45], Catherine M. Tangen[45], Jyotsna Batra[46,47], Judith Clements[46,47], Henrik Gronberg[44], Johanna Schleutker[48,49,50], Demetrius Albanes[43], Stephanie Weinstein[43], Alicja Wolk[51], Catharine West[52], Lorelei Mucci[53], Géraldine Cancel-Tassin[54,55], Stella Koutros[43], Karina Dalsgaard Sorensen[56,57], Eli Marie Grindedal[58], David E. Neal[59,60], Freddie C. Hamdy[61,62], Jenny L. Donovan[63], Ruth C. Travis[64], Robert J. Hamilton[65], Sue Ann Ingles[1], Barry Rosenstein[66,67], Yong-Jie Lu[68], Graham G. Giles[69,70], Adam S. Kibel[71], Ana Vega[72], Manolis Kogevinas[73,74,75,76], Kathryn L. Penney[77], Jong Y. Park[78], Janet L. Stanford[79,80], Cezary Cybulski[81], Børge G. Nordestgaard[82,83], Hermann Brenner[84,85,86], Christiane Maier[87], Jeri Kim[88], Esther M. John[89,90], Manuel R. Teixeira[91,92], Susan L. Neuhausen[93], Kim De Ruyck[94], Azad Razack[95], Lisa F. Newcomb[79,96], Davor Lessel[97], Radka Kaneva[98], Nawaid Usmani[99,100], Frank Claessens[101], Paul A. Townsend[102], Manuela Gago-Dominguez[103,104], Monique J. Roobol[105], Florence Menegaux[106], Kay-Tee Khaw[107], Lisa Cannon-Albright[108,109], Hardev Pandha[110] & Stephen N. Thibodeau[110]

[39]Department of Preventive Medicine, Keck School of Medicine, University of Southern California/Norris Comprehensive Cancer Center, Los Angeles 90033 CA, USA. [40]Department of Epidemiology and Biostatistics, Case Western Reserve University, Cleveland 44106 OH, USA. [41]Seidman Cancer Center, University Hospitals, Cleveland 44106 OH, USA. [42]Department of Clinical Neurosciences, University of Cambridge, Cambridge CB2 1TN, UK. [43]Division of Cancer Epidemiology and Genetics, National Cancer Institute, NIH, Bethesda 20892 MD, USA. [44]Department of Medical Epidemiology and Biostatistics, Karolinska Institute, Stockholm SE-177 77, Sweden. [45]Epidemiology Research Program, American Cancer Society, 250 Williams Street, Atlanta 30303 GA, USA. [46]Australian Prostate Cancer Research Centre-Qld, Institute of Health and Biomedical Innovation and School of Biomedical Science, Queensland University of Technology, Brisbane 4059 QLD, Australia. [47]Translational Research Institute, Brisbane 4102 QLD, Australia. [48]Department of Medical Biochemistry and Genetics, Institute of Biomedicine, University of Turku, Turku FI-20520, Finland. [49]Tyks Microbiology and Genetics, Department of Medical Genetics, Turku University Hospital, Turku FI-20520, Finland. [50]BioMediTech, University of Tampere, Tampere 33100, Finland. [51]Division of Nutritional Epidemiology, Institute of Environmental Medicine, Karolinska Institutet, Solna SE-171 77, Sweden. [52]Institute of Cancer Sciences, University of Manchester, Manchester Academic Health Science Centre, Radiotherapy Related Research, The Christie Hospital NHS Foundation Trust, Manchester M13 9NT, UK. [53]Department of Epidemiology, Harvard School of Public Health, Boston 02115 MA, USA. [54]CeRePP, Pitie-Salpetriere Hospital, Paris 75013, France. [55]UPMC Univ Paris 06, GRC No. 5 ONCOTYPE-URO, CeRePP, Tenon Hospital, Paris 75020, France. [56]Department of Molecular Medicine, Aarhus University Hospital, Aarhus 8000, Denmark. [57]Department of Clinical Medicine, Aarhus University, Aarhus 8000, Denmark. [58]Department of Medical Genetics, Oslo University Hospital, Oslo N-0424, Norway. [59]University of Cambridge, Department of Oncology, Addenbrooke's Hospital, Cambridge CB2 0QQ, UK. [60]Cancer Research UK Cambridge Research Institute, Li Ka Shing Centre, Cambridge CB2 0RE, UK. [61]Nuffield Department of Surgical Sciences, University of Oxford, Oxford OX3 9DU, UK. [62]Faculty of Medical Science, University of Oxford, John Radcliffe Hospital, Oxford OX3 9DU, UK. [63]School of Social and Community Medicine, University of Bristol, Bristol BS8 2PS, UK. [64]Cancer Epidemiology, Nuffield Department of Population Health, University of Oxford, Oxford OX3 7LF, UK. [65]Department of Surgical Oncology, Princess Margaret Cancer Centre, Toronto M5G 2M9, Canada. [66]Department of Radiation Oncology, Icahn School of Medicine at Mount Sinai, New York 10029 NY, USA. [67]Department of Genetics and Genomic Sciences, Icahn School of Medicine at Mount Sinai, New York, NY 10029, USA. [68]Centre for Molecular Oncology, Barts Cancer Institute, Queen Mary University of London, John Vane Science Centre, London EC1M 6BQ, UK. [69]Cancer Epidemiology & Intelligence Division, The Cancer Council Victoria, Melbourne 3004 VIC, Australia. [70]Centre for Epidemiology and Biostatistics, Melbourne School of Population and Global Health, The University of Melbourne, Melbourne 3053 VIC, Australia. [71]Division of Urologic Surgery, Brigham and Womens Hospital, Boston 02115 MA, USA. [72]Fundación Pública Galega de Medicina Xenómica-SERGAS, Grupo de Medicina Xenómica, CIBERER, IDIS, Santiago de Compostela 15782, Spain. [73]Centre for Research in Environmental Epidemiology (CREAL), Barcelona Institute for Global Health (ISGlobal), Barcelona 60803, Spain. [74]CIBER Epidemiología y Salud Pública (CIBERESP), Madrid 28029, Spain. [75]IMIM (Hospital del Mar Research Institute), Barcelona 08003, Spain. [76]Universitat Pompeu Fabra (UPF), Barcelona 08002, Spain. [77]Channing Division of Network Medicine, Department of Medicine, Brigham and Women's Hospital/Harvard Medical School, Boston 02115 MA, USA. [78]Department of Cancer Epidemiology, Moffitt Cancer Center, Tampa 33612 FL, USA. [79]Division of Public Health Sciences, Fred Hutchinson Cancer Research Center, Seattle 98109 WA, USA. [80]Department of Epidemiology, School of Public Health, University of Washington, Seattle 98195 WA, USA. [81]International Hereditary Cancer Center, Department of Genetics and Pathology, Pomeranian Medical University, Szczecin 70-001, Poland. [82]Faculty of Health and Medical Sciences, University of Copenhagen, Copenhagen 1165, Denmark. [83]Department of Clinical Biochemistry, Herlev and Gentofte Hospital, Copenhagen University Hospital, Herlev 2900, Denmark. [84]Division of Clinical Epidemiology and Aging Research, German Cancer Research Center (DKFZ), Heidelberg 69120, Germany. [85]German Cancer Consortium (DKTK), German Cancer Research Center (DKFZ), Heidelberg 69120, Germany. [86]Division of Preventive Oncology, German Cancer Research Center (DKFZ) and National Center for Tumor Diseases (NCT), Heidelberg 69120, Germany. [87]Institute for Human Genetics, University Hospital Ulm, Ulm 89081, Germany. [88]Department of Genitourinary Medical Oncology, The University of Texas M. D. Anderson Cancer Center, Houston 77030 TX, USA. [89]Cancer Prevention Institute of California, Fremont 94538 CA, USA. [90]Department of Health Research & Policy (Epidemiology) and Stanford Cancer Institute, Stanford University School of Medicine, Stanford 94305 CA, USA. [91]Department of Genetics, Portuguese Oncology Institute of Porto, Porto 4200-072, Portugal. [92]Biomedical Sciences Institute (ICBAS), University of Porto, Porto 4200-072, Portugal. [93]Department of Population Sciences, Beckman Research Institute of the City of Hope, Duarte 91016 CA, USA. [94]Ghent University, Faculty of Medicine and Health Sciences, Basic Medical Sciences, Ghent 9000, Belgium. [95]Department of Surgery, Faculty of Medicine, University of Malaya, Kuala Lumpur 50603, Malaysia. [96]Department of Urology, University of Washington, Seattle 98105 WA, USA. [97]Institute of Human Genetics, University Medical Center Hamburg-Eppendorf, Hamburg 20246, Germany. [98]Molecular Medicine Center, Department of Medical Chemistry and Biochemistry, Medical University, Sofia 1431, Bulgaria. [99]Department of Oncology, Cross Cancer Institute, University of Alberta, Edmonton T6G 2R3 Alberta, Canada. [100]Division of Radiation Oncology, Cross Cancer Institute, Edmonton T6G 1Z2 AB, Canada. [101]Molecular Endocrinology Laboratory, Department of Cellular and Molecular Medicine, KU Leuven, Leuven 3000, Belgium. [102]Institute of Cancer Sciences, Manchester Cancer Research Centre, University of Manchester, Manchester Academic Health Science Centre, St. Mary's Hospital, Manchester M13 9WL, UK. [103]Genomic Medicine Group, Galician Foundation of Genomic Medicine, Instituto de Investigacion Sanitaria de Santiago de Compostela (IDIS), Complejo Hospitalario Universitario de Santiago, Servicio Galego de Saúde, SERGAS, Santiago De Compostela 15706, Spain. [104]University of California San Diego, Moores Cancer Center, La Jolla 92093 CA, USA. [105]Department of Urology, Erasmus University Medical Center, Rotterdam 3015, The Netherlands. [106]Cancer & Environment Group, Center for Research in Epidemiology and Population Health (CESP), INSERM, University Paris-Sud, University Paris-Saclay, Villejuif F-94805, France. [107]Clinical Gerontology Unit, University of Cambridge, Cambridge CB2 2QQ, UK. [108]Division of Genetic Epidemiology, Department of Medicine, University of Utah School of Medicine, Salt Lake City 84132 UT, USA. [109]George E. Wahlen Department of Veterans Affairs Medical Center, Salt Lake City 84148 UT, USA. [110]The University of Surrey, Guildford, Surrey GU2 7XH, UK. Deceased: Brian E. Henderson.

