## [Peer Review File · Nature Communications]

Reviewers' comments:

Reviewer #1 (Remarks to the Author):

This manuscript performed the largest GWAS of multiple myeloma to date (9974 cases and 247,556 controls) and have identified six new susceptibility loci for a total of 23. The authors provided biological interpretation of the findings based on gene expression and epigenomic analyses from tumor cells and cell lines, along with in silico analyses. Methods, study designs, and approaches are appropriate and well done. The findings are significant and add to the body of literature about inherited risk for MM.

A few minor comments. The study design was difficult to untangle without effort digging around in the supplementary tables; it wasn't until supplementary table 7, that one found the details of the previous GWAS. Figure 1 was not helpful. Instead, I suggest modifying Figure 1 to include details of the previous GWAS studies along with the new Oncoarray data. The additional detail needed (in addition to sample size) should include the studies from where the samples were recruited. Likewise for "the replication study".

In general, the secondary analyses on subtype-specific associations are based on small numbers. First, provide the total sample size the each meta was based one. Second, the authors should note in the text that their finding of "no evidence for associations" could just be due to small numbers.

Reviewer #2 (Remarks to the Author):

This study represents a meta-analysis of datasets from 3 groups of genome-wide association studies, namely (a) a new set of 878 MM cases and 7,083 controls from the UK (evaluated with the OncoArray platform), (b) 6 previously published MM GWAS data sets (with a total of 7,319 cases and 234,385 controls) and (c) another 1,777 cases and 6,088 controls from three independent series, from Germany, Denmark and Sweden. This study concludes that this meta-analysis identified 6 additional loci associated with myeloma, leading to a total number of 23 loci identified by the GWAS studies and meta-analyses of the current group of authors. This current study also incorporates a re-analysis of in situ promoter capture Hi-C (CHi-C) data (which the authors had generated in a previous paper) in a MM cell line and publicly available data ChIP-seq data for several histone marks on a lymphoblastoid B cell line and a MM cell line. The goal of these analyses was to examine potential associations of identified myeloma-related loci/SNPs with known or presumed myeloma-related genes through chromatin looping interactions.

Major comments:

1) A general comment that permeates this entire review (and several of the points below): the manuscript would greatly benefit from defining more clearly why it is distinct from and a major advancement compared to the previous Nature Communications (2016) paper of the same group; as well as the strengths and additional features of this study compared to prior attempts to meta-analyze GWAS data in MM.

2) What is the explanation for the identification of the 6 new loci in this meta-analysis, but not prior ones?

The current manuscript involves a meta-analysis of a total 9,974 MM cases and 247,556 controls, compared to 9,866 cases and 239,188 controls in the 2016 Nature Communications paper from a similar group of authors. Since these numbers are not very different, why did this current study found 6 additional loci compared to the 17 identified two years ago? Can this be a matter of different statistical power, when the studies have similar overall size? Is this due to heterogeneity of the studies included in this new meta-analysis vs. the previous one?

3) How big is the contribution of these 6 additional chromosomal locations to the attributable genetic heritability of myeloma?

Page 6 states the 23 (17 + 6) new and previously identified risk variants identified from the current study locations represent 20% of the attributable genetic heritability of MM, but it is not clear what part of this % attributable to the 6 new loci vs. the previous 17 ones.

4) On a related question, it is important for this study to attempt to provide an estimate, based on its current data, on how many more cases and controls might be needed in order to improve our understanding of the genetic heritability of myeloma to much a higher % than 20%. It is important that this study attempts to not only describe the current results, but also give readers an opportunity to appreciate in a data-driven manner, if GWAS studies have actually reached a plateau in their ability

to identify new variants with major contribution to myeloma risk, or if additional GWAS studies can still improve our understanding of genetic heritability of this disease, and how large the number of aggregate cases and controls should be in order to reach different % of desirable attribution of genetic heritability of MM. Please note that such "saturation" analyses have been conducted for e.g. next-generation sequencing of patient-derived tumor samples to inform how many samples from different tumors will be needed to comprehensively describe their mutational landscape. It is hard to imagine that similar "saturation" analyses are not possible for GWAS studies in MM and it is in fact critical for the field and the authors themselves to have those analyses done so that the MM field has an idea of what to realistically expect from its future GWAS efforts.

5) The results of Table 2, supplemental figure 5 and supplemental table 13, and the overall effort of the paper to link the new disease-associated loci with potential transcription factor binding sites in the loci is too descriptive and rather speculative. Unless these sections are thoroughly modified and improved, it is probably better to even remove them. According to the authors' own data, the sites containing the disease-associated loci typically contain open chromatin in cells of the B-cell lineage, so it is perfectly reasonable to expect that diverse transcription factors would be binding in these regions. However, the GM12878 cells are not myeloma cells, but B lymphoblastoid cells and the several of the "B-cell relevant TF binding" data shown on this graph may not be relevant to myelomagenesis. Many, if not most, of the TFs listed in table 2 or supplemental table 13 do not have a specific role in normal or malignant plasma cells, and there is also limited information from the paper to suggest that there is actual chromatin occupancy in these sites by the respective transcription factors. Please also note that several of the genes mentioned in the paper as being close to the identified risk loci (e.g. SP3, PRR14) actually have very tenuous, if any, actual link with MM pathophysiology.

6) It would be a great benefit to the study and the field, if there is more emphasis on comparing rigorously the results of the current study with others (especially the ones whose data were not incorporated in the current GWAS meta-analysis), particularly for genes/loci which may have come up as significant in other studies, but not this one (or vice versa) and potential reasons (e.g. differences between different GWAS studies) for these contrasting results.

7) The paper would also benefit from providing to readers (e.g. in the supplement) more information on the interpretation of some quantitative parameters provided in the results of these GWAS and their meta-analyses. For example, the definition and significance of the I2 value in Table 1 or Supplemental Table 4; and of the P(HET) values (Supplemental Table 4) should be defined more clearly.

8) Supplemental Figure 3: It will likely be unclear to readers who are not GWAS experts how big or small are the differences in polygenic risk scores for familial vs. Sporadic MM vs. population controls. It is recommended that the authors provide in the legend (or supplement) some form of measure of which level of quantitative differences in PRS and p-values have been detected in studies of other malignancies with major (or conversely, minor) differences in the biology and heritable component of their familial vs. sporadic cases.

9) Supplemental Figure 6: The removal from this study of cases with non-european ancestry suggests that the title and/or abstract of the study should highlight that this study was restricted to Caucasian populations, so that it is clear to a reader that results are potentially distinct from past, ongoing or future GWAS studies on non-Caucasian populations.

10) Supplementary Table 1: Please comment on the choice of p-value cutoff ($P < 1.0 \times 10^{-6}$) for the meta-analyses (MetaGWAS or MetaGWAS+REP): GWAS meta-analyses in other cancers have often used lower p-value cutoffs, so it would be useful to justify this choice here.

Minor comments:

11) As a general comment, the paper is written as if it were going to be read by GWAS experts within the same disease or others but not towards a broader audience that doesn't have as much familiarity with the results of these studies and their interpretation. It would help the paper, especially in a journal such as Nature Communications, to provide, either in the main paper or the supplement, a context for some of these results, e.g. explain to a non-expert what the different statistical tests mean, which values for their metrics represent significant vs. borderline vs. non-significant observations, et.c.

12) Supplementary Table 2: Many of the p-values from the individual validation studies seem to be >0.05 in the different validation cohorts examined. The interpretation of these findings should be discussed. If the authors interpret these results to mean that these studies contribute to the "validation" of a given loci only through their participation in the meta-analyses, this should be clarified.

Reviewer #1:

This manuscript performed the largest GWAS of multiple myeloma to date (9974 cases and 247,556 controls) and have identified six new susceptibility loci for a total of 23. The authors provided biological interpretation of the findings based on gene expression and epigenomic analyses from tumor cells and cell lines, along with in silico analyses. Methods, study designs, and approaches are appropriate and well done. The findings are significant and add to the body of literature about inherited risk for MM.

1.1) *The study design was difficult to untangle without effort digging around in the supplementary tables; it wasn't until supplementary table 7, that one found the details of the previous GWAS. Figure 1 was not helpful. Instead, I suggest modifying Figure 1 to include details of the previous GWAS studies along with the new Oncoarray data. The additional detail needed (in addition to sample size) should include the studies from where the samples were recruited. Likewise, for "the replication study".*

Response: Fig. 1 has been modified to now provide information on each of the 6 previously published GWAS, including their sample size, together with the new GWAS. The trials or recruitment centres from which the samples were obtained are included in Supplementary Table 1. Similarly, for the replication samples details of where these samples were obtained are included in Supplementary Table 3.

1.2) *In general, the secondary analyses on subtype-specific associations are based on small numbers. First, provide the total sample size the each meta was based one. Second, the authors should note in the text that their finding of "no evidence for associations" could just be due to small numbers.*

Response: The table legends for Supplementary Tables 5a-d and Supplementary Table 6 (now Supplementary Tables 7a-d and 8, respectively) have been modified to include information on the total sample size for the meta-analysis. The manuscript text has been revised. Specifically, we now state: "Aside from previously reported relationship between the risk loci at 11q13.3 and 5q15 with t(11;14) MM and hyperdiploid MM respectively, we found no evidence for subtype-specific associations for other risk SNPs (**Supplementary Table 7a-d**) or an impact on MM-specific survival (**Supplementary Table 8**). A failure to demonstrate additional relationships may however be reflective of limited study power."

Reviewer #2:

This study represents a meta-analysis of datasets from 3 groups of genome-wide association studies, namely (a) a new set of 878 MM cases and 7,083 controls from the UK (evaluated with the OncoArray platform), (b) 6 previously published MM GWAS data sets (with a total of 7,319 cases and 234,385 controls) and (c) another 1,777 cases and 6,088 controls from three independent series, from Germany, Denmark and Sweden. This study concludes that this meta-analysis identified 6 additional loci associated with myeloma, leading to a total number of 23 loci identified by the GWAS studies and meta-analyses of the current group of authors. This current study also incorporates a re-analysis of in situ promoter capture Hi-C (ChIP-C) data (which the authors had generated in a previous paper) in a MM cell line and publicly available data ChIP-seq data for several histone marks on a lymphoblastoid B cell line and a MM cell line. The goal of these analyses was to examine potential associations of identified myeloma-related loci/SNPs with known or presumed myeloma-related genes through chromatin looping interactions.

2.1) *A general comment that permeates this entire review (and several of the points below): the manuscript would greatly benefit from defining more clearly why it is distinct from and a major advancement compared to the previous Nature Communications (2016) paper of the same group; as well as the strengths and additional features of this study compared to prior attempts to meta-analyze GWAS data in MM.*

Response: This study reports a new GWAS and meta-analysis of six previously published GWAS. We additionally state that we used Capture Hi-C (CHI-C) and chromatin immunoprecipitation-sequencing (ChIP-seq) to annotate 23 MM risk loci and identify target genes. To assess the support for these target genes in other data sources we tested for associations between levels of expression and SNP genotype (eQTL). The manuscript text has been modified to emphasise these salient points.

2.2) *What is the explanation for the identification of the 6 new loci in this meta-analysis, but not prior ones?*

The current manuscript involves a meta-analysis of a total 9,974 MM cases and 247,556 controls, compared to 9,866 cases and 239,188 controls in the 2016 Nature Communications paper from a similar group of authors. Since these numbers are not very different, why did this current study found 6 additional loci compared to the 17 identified two years ago? Can this be a matter of different statistical power, when the studies have similar overall size? Is this due to heterogeneity of the studies included in this new meta-analysis vs. the previous one?

Response: While numbers may look similar, the Mitchell *et al* paper was based on 7,319 cases for discovery. Our current paper, which includes an additional 878 cases provides for a discovery series of 8,197. Adopting the two stage strategy the current study provided around 10% increased power to identify new loci. To the best of our knowledge there is no salient difference between the clinico-pathology of cases analysed in the new GWAS as compared to the published series. Of note there was no significant heterogeneity (P_{HET} calculated for Cochran's Q statistic and the I^2 values) for each of the risk SNPs we identified.

2.3) *How big is the contribution of these 6 additional chromosomal locations to the attributable genetic heritability of myeloma?*

Page 6 states the 23 (17 + 6) new and previously identified risk variants identified from the current study locations represent 20% of the attributable genetic heritability of MM, but it is not clear what part of this % attributable to the 6 new loci vs. the previous 17 ones.

Response: Information on the heritability ascribed to the new and old loci has been calculated and this information is now provided in the revised text.

2.4) *On a related question, it is important for this study to attempt to provide an estimate, based on its current data, on how many more cases and controls might be needed in order to improve our understanding of the genetic heritability of myeloma to much a higher % than 20%. It is important that this study attempts to not only describe the current results, but also give readers an opportunity to appreciate in a data-driven manner, if GWAS studies have actually reached a plateau in their ability to identify new variants with major contribution to myeloma risk, or if additional GWAS studies can still improve our understanding of genetic heritability of this disease, and how large the number of aggregate cases and controls should be in order to reach different % of desirable attribution of genetic heritability of MM. Please note that such "saturation" analyses have been conducted for e.g. next-generation sequencing of patient-derived tumor samples to inform how many samples from different tumors will be needed to comprehensively describe their mutational landscape. It is hard to imagine that similar "saturation" analyses are not possible for GWAS studies in MM and it is in fact critical for the field and the authors themselves to have those analyses done so that the MM field has an idea of what to realistically expect from its future GWAS efforts.*

Response: To estimate sample size required we implemented a likelihood-based approach using association statistics and LD information modelling effect-size distributions as per Chatterjee *et al*^{1,2}. This information and results from the analysis are now included in the revised text.

2.5) *The results of Table 2, supplemental figure 5 and supplemental table 13, and the overall effort of the paper to link the new disease-associated loci with potential transcription factor binding sites in the loci is too descriptive and rather speculative. Unless these sections are thoroughly modified and improved, it is probably better to even remove them. According to the authors' own data, the sites containing the disease-associated loci typically contain open chromatin in cells of the B-cell lineage, so it is perfectly reasonable to expect that diverse transcription factors would be binding in these regions. However, the GM12878 cells are not myeloma cells, but B lymphoblastoid cells and the several of the "B-cell relevant TF binding" data shown on this graph may not be relevant to myelomagenesis. Many, if not most, of the TFs listed in table 2 or supplemental table 13 do not have a specific role in normal or malignant plasma cells, and there is also limited information from the paper to suggest that there is actual chromatin occupancy in these sites by the respective transcription factors. Please also note that several of the genes mentioned in the paper as being close to the identified risk loci (e.g. SP3, PRR14) actually have very tenuous, if any, actual link with MM pathophysiology.*

Response: We cannot infer at what stage in B-cell development any of the risk SNPs are likely to impact on disease risk. Therefore we would assert it is logical to examine different stages of B-cell/ plasma cell maturation. We have included GM12878 lymphoblastoid cells as a representative model of a normal B-cell, while our analysis of incorporation of the KMS11 cell line, provides represents a model of MM. We have sought to incorporate primary cells alongside cell lines in the form of patient expression data from studies included in the GWAS, as well as publically available naïve B cell data, again representing an early, or possibly, 'pre-disease' state. By including both early and late disease models in the form of cell lines and, in addition, primary cells, we hope to capture an understanding of both MM pathophysiology and pathogenesis. Furthermore, germline variants need not act on B-cells; it is plausible, for example, that a variant may act in a tissue of the microenvironment and thus influence the trajectory of MM.

We have incorporated data on transcription factor binding, alongside ChIP-seq data, as this provides orthogonal evidence to implicate these loci as candidate genes in MM pathogenesis.

The genes identified in the current GWAS provide novel associations for MM, so it is not expected that there should be any link to MM pathophysiology, though intriguingly, one of the candidate genes, *KLF2*, had a detailed study of its role in MM pathophysiology. Moreover, genes such as the specified transcription factor, *SP3*, have a role B-cell development, specifically in this case the regulation of germinal centre genes, hence they provide a biologically plausible gene of interest for MM pathogenesis.

2.6) *It would be a great benefit to the study and the field, if there is more emphasis on comparing rigorously the results of the current study with others (especially the ones whose data were not incorporated in the current GWAS meta-analysis), particularly for genes/loci which may have come up as significant in other studies, but not this one (or vice versa) and potential reasons (e.g. differences between different GWAS studies) for these contrasting results.*

Response: To our knowledge our analysis is the largest GWAS of MM reported to date. The report by Rand *et al* a GWAS of individuals of African ancestry and European ancestry only assessed loci we previously discovered in Broderick *et al*³ (Nature Genetics, 2011) and Chubb *et al*⁴ (Nature Genetics, 2013), specifically, SNPs at 7 loci. Additionally they assessed the putative risk locus at 2q12.3 for MM reported by Erickson *et al*⁵ (Blood, 2014). This association was not, however, genome-wide significant in the report since "replication analysis" of the GWAS signal was solely based on an analysis of cases with no independent controls. We have expanded our text to include reference and commentary of these reports.

2.7) *The paper would also benefit from providing to readers (e.g. in the supplement) more information on the interpretation of some quantitative parameters provided in the results of these GWAS and their meta-analyses. For example, the definition and significance of the I^2 value in Table 1 or Supplemental Table 4; and of the $P(HET)$ values (Supplemental Table 4) should be defined more clearly.*

Response: The manuscript has been revised to now define the significance of the I^2 and P_{HET} values in the supplementary table legend, in addition to in the methods.

2.8) *Supplemental Figure 3: It will likely be unclear to readers who are not GWAS experts how big or small are the differences in polygenic risk scores for familial vs. Sporadic MM vs. population controls. It is recommended that the authors provide in the legend (or supplement) some form of measure of which level of quantitative differences in PRS and p-values have been detected in studies of other malignancies with major (or conversely, minor) differences in the biology and heritable component of their familial vs. sporadic cases.*

Response: We now provide commentary on this in the text and Supplementary Figure legend.

2.9) *Supplemental Figure 6: The removal from this study of cases with non-european ancestry suggests that the title and/or abstract of the study should highlight that this study was restricted to Caucasian populations, so that it is clear to a reader that results are potentially distinct from past, ongoing or future GWAS studies on non-Caucasian populations.*

Response: The abstract has been modified to include state that this study was conducted in individuals of European ancestry.

2.10) *Supplementary Table 1: Please comment on the choice of p-value cutoff ($P < 1.0 \times 10^{-6}$) for the meta-analyses (MetaGWAS or MetaGWAS+REP): GWAS meta-analyses in other cancers have often used lower p-value cutoffs, so it would be useful to justify this choice here.*

Response: The significance threshold used to identify variants which showed evidence of association is dependent on the size of GWAS and of the replication size available. Given the large discovery GWAS and modest replication size $P < 5 \times 10^{-6}$ was chosen to maximise the potential of discovering true associations in the replication stage. In this study we adopt a two stage strategy, which provides around 10% increased power to identify new loci. To be considered genome-wide significant and a *bona fide* MM GWAS risk locus an association in the meta-analysis of the GWAS and replication datasets with significance level $P < 5 \times 10^{-8}$ was required. Supplementary Table 4 legend has been modified to clarify this.

Minor comments:

2.11) *As a general comment, the paper is written as if it were going to be read by GWAS experts within the same disease or others but not towards a broader audience that doesn't have as much familiarity with the results of these studies and their interpretation. It would help the paper, especially in a journal such as Nature Communications, to provide, either in the main paper or the supplement, a context for some of these results, e.g. explain to a non-expert what the different statistical tests mean, which values for their metrics represent significant vs. borderline vs. non-significant observations, et.c.*

Response: We have revised the text to clarify what constitutes genome-wide significance and provided an appropriate reference.

2.12) *Supplementary Table 2: Many of the p-values from the individual validation studies seem to be >0.05 in the different validation cohorts examined. The interpretation of these findings should be discussed. If the*

authors interpret these results to mean that these studies contribute to the "validation" of a given loci only through their participation in the meta-analyses, this should be clarified.

Response: The Swedish and Danish replication series had smaller sample sizes than those in the GWAS datasets, hence individually they are not necessarily powered to demonstrate an association $P < 0.05$. The main point is that that collectively they support associations as point estimates for the associations were consistent.

Yours Sincerely,

Dr Richard S Houlston MD PhD DSc FMedSci
Professor in Molecular and Population Genetics

References

1. Chatterjee, N. *et al.* Projecting the performance of risk prediction based on polygenic analyses of genome-wide association studies. *Nat Genet* **45**, 400-5, 405e1-3 (2013).
2. Chatterjee, N. Estimation of complex effect-size distributions using summary-level statistics from genome-wide association studies across 32 complex traits and implications for the future. *BioRxiv* (2017).
3. Broderick, P. *et al.* Common variation at 3p22.1 and 7p15.3 influences multiple myeloma risk. *Nat Genet* **44**, 58-61 (2011).
4. Chubb, D. *et al.* Common variation at 3q26.2, 6p21.33, 17p11.2 and 22q13.1 influences multiple myeloma risk. *Nat Genet* **45**, 1221-1225 (2013).
5. Erickson, S.W. *et al.* Genome-wide scan identifies variant in 2q12.3 associated with risk for multiple myeloma. *Blood* **124**, 2001-3 (2014).

REVIEWERS' COMMENTS:

Reviewer #1 (Remarks to the Author):

The authors addressed my comments. I have no other concerns.

Reviewer #2 (Remarks to the Author):

The revised version of this manuscript addresses comprehensively the comments of my review.